# Engineering Smooth Muscle to Understand Extracellular Matrix Remodeling and Vascular Disease

**DOI:** 10.3390/bioengineering9090449

**Published:** 2022-09-07

**Authors:** Danielle Yarbrough, Sharon Gerecht

**Affiliations:** 1Department of Biomedical Engineering, Duke University, Durham, NC 27708, USA; 2Department of Chemical and Biomolecular Engineering, Johns Hopkins University, Baltimore, MD 21218, USA

**Keywords:** tissue engineering, extracellular matrix, vascular smooth muscle cells, cardiovascular disease

## Abstract

The vascular smooth muscle is vital for regulating blood pressure and maintaining cardiovascular health, and the resident smooth muscle cells (SMCs) in blood vessel walls rely on specific mechanical and biochemical signals to carry out these functions. Any slight change in their surrounding environment causes swift changes in their phenotype and secretory profile, leading to changes in the structure and functionality of vessel walls that cause pathological conditions. To adequately treat vascular diseases, it is essential to understand how SMCs crosstalk with their surrounding extracellular matrix (ECM). Here, we summarize in vivo and traditional in vitro studies of pathological vessel wall remodeling due to the SMC phenotype and, conversely, the SMC behavior in response to key ECM properties. We then analyze how three-dimensional tissue engineering approaches provide opportunities to model SMCs’ response to specific stimuli in the human body. Additionally, we review how applying biomechanical forces and biochemical stimulation, such as pulsatile fluid flow and secreted factors from other cell types, allows us to study disease mechanisms. Overall, we propose that in vitro tissue engineering of human vascular smooth muscle can facilitate a better understanding of relevant cardiovascular diseases using high throughput experiments, thus potentially leading to therapeutics or treatments to be tested in the future.

## 1. Introduction

Cardiovascular disease (CVD) is the leading cause of death worldwide and involves a myriad of conditions evolved from different disease states affecting the behavior of the major cell types in the vascular system [1]. Vascular smooth muscle cells (SMCs) play an integral role in vasoconstriction and blood pressure regulation, and the pathological dysregulation of these vascular properties due to SMC dysfunction can lead to life-threatening conditions. Along with fibroblasts, SMCs secrete and degrade the vessel wall’s extracellular matrix (ECM). The components, structure, and mechanical properties of the ECM dictate SMC phenotype via biomechanical and biochemical signals, and the SMCs, in turn, remodel the ECM via secretory factors and direct interaction via the focal adhesions [2]. These interactions are essential to understand how the ECM’s crosstalk with the vasculature contributes to life-threatening disease states, many of which are brought on by stiffening arterial walls. 

Animal models help us to understand these disease states on a systemic level [3]. The genome of rodents can be altered so that a specific phenotype corresponding to a disease state of interest can be studied. However, there are reported differences in the physiology and overall functionality of human vasculature compared to animal models [4], and induced genetic mutations in rodents may not necessarily result in a relevant phenotype in humans suffering from a similar genetic defect. Large animal models, such as swine and sheep, allow for the pre-clinical testing of potential surgical or therapeutic treatments for severe cardiovascular events [5]. Nonetheless, extensive animal studies are expensive and not amenable to high throughput experimentation. 

Tissue engineering with human cells offers a unique opportunity and ability to model biological cues accurately in vitro [6]. Thus, engineered tissues can better recapitulate how cells are likely to behave in the human body and enable a better understanding of diseases and the underlying mechanisms governing them. In this review, we discuss recent studies using traditional in vitro culture systems and animal models that have improved our understanding of how specific properties of the ECM in the vasculature modulate the SMC phenotype, and how the SMCs, in turn, alter the makeup of that ECM. We also discuss how other external cues, such as flow, mechanical strain, and biochemical signals, are used to create three-dimensional (3D) tissue engineered in vitro models of vasculature. These models can help further elucidate the SMC behavior and contribute to a more profound knowledge of cardiovascular diseases resulting from smooth muscle tissue dysfunction. 

## 2. ECM Properties Affecting SMC Phenotype

Because SMCs are so vital to healthy vascular function, there is an extensive base of knowledge around the various CVD phenotypes related to SMC behavior. Many of these disease phenotypes result from SMCs exhibiting phenotypic plasticity, meaning they can shift between a more mature or immature phenotype, depending on signals from the ECM. When SMCs are in a more immature state seen during development and vascular disease or injury, known as the synthetic phenotype, they proliferate quickly and secrete more significant amounts of ECM proteins. On the other hand, when the vessels are healthy, SMCs are typically in a more mature state, known as the contractile phenotype [7]. In vivo studies, particularly in rodent models, have pinpointed the specific ECM molecules’ role in SMC phenotypic shifting and the overall vascular function resulting from this behavior. The ability to genetically modify these organisms has given investigators insight into how the entire system is affected by the increased expression of a signaling molecule or the absence of a certain protein. The isolation and in vitro culture of SMCs from these in vivo models or human tissues has additionally prompted further insight into some of the mechanisms behind these changes. 

When SMCs are cultured on 2D substrates, detailed phenotypic characteristics can be evaluated through a variety of methods. Contractile marker expression can be quantified as protein or RNA transcript [8,9]. The degree of functional contractility can be determined via planar cell surface area (PCSA) measurement before and after the treatment of cells with vasodilating or vasoconstricting agents [10,11,12]. The calcium and potassium ion channel activity, which directly induces the SMC contractile function, can be evaluated via electrical current measurement across the cell membrane using patch clamp electrophysiology [13,14,15]. Recent studies using methods such as these and the major findings are discussed below and summarized in Figure 1 and Table 1. 

### 2.1. Stiffness

Stiffness is an important mechanical property of the ECM in vessel walls and tightly regulates the ability of SMCs to control blood pressure. In turn, increased ECM stiffness, associated with many prominent CVDs, can be exacerbated from changes in the expression of specific genes and secreted factors by the cells. Genetically modified mice have been used extensively to identify the mediators of vascular ECM stiffness in SMCs. In a study of age-related vascular stiffening, the depletion of lysyl oxidase-like 2 (LOXL2) results in increased stiffening with age in mice [16], and mineralocorticoid receptor depletion in aged mice and pharmacological inhibition in elderly humans successfully mitigates vascular fibrosis [17]. In addition, young mice introduced to blood flow from old mice via parabiosis show an upregulation of the genes related to the pathologic vascular wall remodeling [18]. Focal adhesion kinases (FAK) and related focal adhesion proteins have also been an area of interest in studying ECM stiffness effects, as SMC focal adhesions are highly dependent on the stiffness of their matrix and regulate the contractile capability of the cells [37]. Transgenic mouse models have revealed that the aberrant activation of FAKs (chemically induced) contributes to increased neointimal hyperplasia via cyclin D1 signaling [9]. Additionally, SMC proliferation and neointima formation were reduced in response to injury when FAK and/or N-cadherin genes are knocked out. Complementary 2D in vitro studies indicated that N-cadherin mediates stronger cell–cell adhesions and increased the proliferation rate in response to increased ECM stiffness and FAK activation [19].

For many in vitro studies, the cells are cultured in two-dimensional (2D) Petri dishes or gels engineered to have tunable chemistry and tightly controlled stiffness properties. This allows for a detailed evaluation of how single variable changes in substrate stiffness or other matrix mechanical properties affect cells. SMCs cultured on collagen I-coated polyacrylamide (PA) gels (stiffness ranging from 1 to 100 kPa) exhibit increased expression of contractile markers on stiffer substrates, mediated through transforming growth factor beta (TGFβ) signaling [8], while SMCs on fibronectin-coated PA gels shifted to a synthetic phenotype through the downregulation of DNA methyltransferase 1 as substrate stiffness increased [22]. The SMCs deposit high levels of calcium and undergo osteogenesis on poly(dimethylsiloxane) (PDMS) substrates of intermediate stiffness (0.91 MPa) when compared to substrates of extreme high (2.33 MPa) or low (0.36 MPa) stiffness [20]. Meanwhile, the actual stiffness of the cytoskeleton in the SMCs can be affected by the properties of the surrounding matrix. Cytoskeletal stiffness can be analyzed using atomic force microscopy (AFM), and this method has been used to show that increased substrate stiffness results in significantly larger traction forces applied by aortic SMCs [23]. Alternatively, an increase in cytoskeletal stiffness brought on by genetic mutation, such as a gain-of function mutation in the hypoxia-inducible factor 2α gene, can lead to a positive feedback loop and signaling cascade, ultimately leading to pathological vessel wall stiffening and hypertension [38]. 

### 2.2. Fibrillar Protein Composition

The composition of the ECM surrounding the SMCs in the vessel walls is tightly regulated throughout the development and maturation of the vasculature. The most abundant components in the ECM (most notably collagen, elastin, fibronectin, and laminin) make up the fibrillar network and work together to provide support and flexibility to blood vessels that need to expand and contract steadily throughout the human lifespan. When the vasculature is diseased, the ratio of these vital components shifts due to changes in ECM protein secretion and expression of ECM modifiers (crosslinkers and proteases) [39]. In a diseased state, SMCs in their synthetic phenotype exhibit increased collagen expression, decreased expression of mature contractile markers, and increased cell migration and proliferation. On the other hand, healthy SMCs in their mature contractile phenotype produce higher levels of elastin, and express robust contractile markers, such as smooth muscle myosin heavy chain (SM-MHC) [7,40].

To understand their relationship with the ECM, these components are used in vitro to promote either contractile or synthetic phenotype in SMCs. For example, the addition of soluble fibronectin in SMC-laden collagen I gel leads to faster gel contraction, enhanced mechanical tension and compression properties, and upregulation of elastin assembly proteins [24]. Additionally, by incorporating increased proportions of insoluble elastin into fibrillar collagen in SMC-laden gels, the contractile marker expression increases [25]. 

The synergistic effects of mechanical and biochemical cues of ECM components are also considered. For example, in SMCs cultured on stiffness-tunable gels with ECM coatings, collagen I promotes a less proliferative and more migratory phenotype with increased stiffness, while gels coated in fibronectin induce the opposite effects [26]. Meanwhile, the myogenic differentiation of mesenchymal stem cells cultured on silk fibroin hydrogels with TGF-β1 supplementation is significantly increased on soft gels (6 kPa) compared with stiff (33 kPa) gels [21].

### 2.3. Non-Fibrillar Proteins and Matrix Modifiers

The importance of matricellular proteins has become increasingly apparent in recent studies as the degree of crosslinking, presence of precursor and chaperone molecules, and the post-translational modifications play a vital yet complex role in determining ECM structural properties [2,41]. Numerous matricellular proteins regulate SMC behavior and overall vascular dysfunction, and studies using isolated human cells or genetically modified rodents have helped to pinpoint many of these proteins. Advanced Glycation End products (AGEs) increase vascular stiffness via the crosslinking of collagen and gene expression modulation via inflammatory cascade [27]. Matrix metalloproteinase (MMP) -12 production is induced in SMCs after vascular injury in mice and is accompanied by increased vascular stiffness, while deletion of MMP-12 in mice abrogates arterial stiffening via a reduction in elastin degradation [35]. The resident macrophages in mouse and human aortic tissue interact with the hyaluronic acid ECM produced by SMCs and can modulate SMC collagen expression via MMP-9 production, ultimately preventing arterial stiffness [36]. The Serum Response Factor (SRF) regulates the SMC phenotype through the expression of a protein phosphatase, PTEN: with decreased PTEN expression in SMCs isolated from human atherosclerotic lesions [34]. RhoBTB1 attenuates vascular stiffness via actin depolymerization in mice with angiotensin-II-induced hypertension, though it does not reverse the hypertension [29]. Meanwhile, disruptions in the polymerization of elastin due to various genetic mutations in mice degrade the integrity of elastin-contractile units with SMCs, which lethally affect the material properties of the arterial wall [30]. 

Again, in vitro studies are used to elucidate certain effects of non-fibrillar proteins that may be difficult to decipher using mice. Tunable gels have been used to show that increased collagen crosslinking induces ECM calcification and osteogenic trans-differentiation in SMCs [32]. Additionally, a hyaluronic-acid micropatterned surface used for culturing vascular cells promotes a contractile phenotype and decreased proliferation rate in SMCs [28]. In an example of the effects of post-translational modifications, SMC adhesion to glycosylated fibronectin is significantly increased when compared to adhesion affinity with native fibronectin [33].

In the interest of clinical applications for these findings, non-invasive methods for studying human vessels, such as pulse wave velocity (PWV) measurements, have been combined with proteome analysis of patient samples. In one study using this method, the matricellular proteins involved in collagen fibril assembly and turnover were significantly downregulated in samples from the patients with high PWV, implicating their role in arterial stiffness and SMC dysfunction [31].

## 3. Engineering Complex In Vitro Models of Smooth Muscle

Traditional cell culture methods and animal models have yielded a vast improvement in the knowledge of how specific aspects of the SMC phenotype are affected by individual changes in the properties and composition of the surrounding matrix. However, it is still a challenge to prompt SMCs to behave in culture as they behave in the body [40], so there is a gap in our understanding of how exactly a specific pathology seen in vivo may be triggered on a cellular and molecular level [42,43]. This gap can be filled by using more complex culture systems incorporating more of the biological cues that SMCs experience in the body. These systems can range from simple mechanical stimulation to co-culture with other cell types and organ-on-a-chip devices to complex tissue engineered vascular grafts, as illustrated in Figure 2. 

### 3.1. Mechanical Stimulation in Culture Systems

Because SMCs are sensitive mechotransducers, many in vitro models attempted to mimic physiological mechanical forces through various methods, such as axial stretch, vacuum-driven strain, radial compression, or interstitial flow, as illustrated in Figure 2. Using these models, investigators have observed that SMCs behave differently under mechanical strain than in traditional static culture. This indicates that adding mechanical cues to culture systems may better encompass the conditions needed to correlate in vitro results with SMC behavior in vivo [43]. 

Cyclic strain is a standard method of mechanical stimulation applied to cell culture constructs to induce the desired SMC behavior [44,45,46]. In one study, incrementally increasing the frequency of cyclic stretch resulted in a higher degree of SMC alignment and denser collagen structure but decreased the expression of many ECM proteins and proteases compared to unstretched controls [47]. The impact of cyclic strain is also directly compared between 2D and 3D SMC cultures. In one example of this, SMCs in 3D constructs align parallel to the direction of strain and shift to a more contractile phenotype, while the cells in 2D align perpendicularly to the direction of strain and showed no changes in contractility [48]. Shear effects have also been compared between 2D and 3D—an increased contractile phenotype was induced in SMCs cultured with interstitial flow in 3D gels compared to SMCs cultured under laminar flow in 2D [49]. Alternatively to cyclic stretch, the centrifugal compression of SMC-laden collagen gels has been used to simulate wall radial strain, and compressed cells exhibited an increased expression of alpha-smooth muscle actin (αSMA) and a decreased expression of MMPs and their inhibitors controls [50]. 

### 3.2. Organ-on-a-Chip and Multicellular Systems

While mechanical stimulation does provide the strain that SMCs would experience natively, the cells may still not respond as they would physiologically if they are not cultured with other tissue cell types that communicate with SMCs and affect their interactions with the surrounding matrix [42]. Thus, more complex models utilizing the 3D environment in combination with co-culture techniques have been used to prompt SMC maturation and more in vivo-like characteristics.

One promising direction for the in vitro modeling of SMC behavior in 3D matrices is organ-on-a-chip technology [51]. Cellular crosstalk was successfully demonstrated between ECs and SMCs when both of the cell types were cultured in a fibrin gel in a microfluidic device modeling vasculogenesis [52]. A versatile device containing four microfluidic parallel channels was used to co-culture SMCs and ECs in an anatomically accurate geometry with tunable channel structures for different flow conditions [53]. Meanwhile, a microfluidic platform was fabricated for the culture and detailed assessment of explanted small arteries [54]. 

Apart from microfluidics and organ-on-a-chip, other co-culture and 3D studies have shown significant changes in cell behavior due to signaling between SMCs and other cell types. The SMCs cultured with ECs in electrospun fibrin microfibers yielded a robust microvascular structure with significant deposition of collagen I and elastin by SMCs [55]. When culturing monocytes with SMCs at different ratios, a 2:1 and 4:1 ratio resulted in balanced protease activity and ECM deposition rate by SMCs, representing healthy tissue [56]. The SMCs cultured on fibrin discs or strands were treated with adipose stromal cell-secreted factors—this induced increased expression of tropoelastin and elastin assembly proteins and increased high stretch modulus, indicating the deposition of mature collagen [57]. The maturation of mesenchymal cells into SMC-like cells has been achieved through co-culture with fibroblasts in decellularized ECM; this maturation was confirmed using the PCSA analysis [11]. Alternatively, endothelial cell adhesion properties were altered when the cells were exposed to media from SMCs exposed to cyclic stretch, indicating the importance of cell signaling from SMCs to other vascular cells as well [58].

### 3.3. Tissue Engineered Vascular Grafts and Pulsatile flow

As the most complex type of in vitro model, tissue engineered vascular grafts provide the opportunity to recapitulate physiological geometry, biomechanical, and biochemical. However, when constructing these systems, little focus is given to the accuracy of the ECM components, structure, and bioactive nature. Instead, investigators prioritize the matrix mechanical properties of strength and elasticity, with the clinical application of these grafts typically an ultimate goal [6,59]. 

The common materials used for vascular grafts include synthetic polymer materials such as poly(glycolic acid) (PGA), polycaprolactone (PCL), and polytetrafluoroethylene (PTFE) [60,61,62,63]. These polymeric materials are specifically chosen for their proven strength, biocompatibility, and elastic properties, as well as a well-characterized slow rate of degradation in the body [64]. These properties have made them an attractive option for clinical applications in recent years. For example, when PGA scaffolds were seeded with human-induced pluripotent stem cell (iPSC)-derived SMCs and subjected to a radial strain of 1% via pulsatile flow for 7 weeks, SMCs produced robust ECM and acquired a contractile phenotype. These matured tissue constructs remained patent for 30 days after implantation into nude rats [60]. Poly(ethylene glycol)l dimethacrylate/poly(L-lactide) scaffolds were seeded with iPSC-derived SMCs and subjected to pulsatile flow through the scaffold lumen for 6 days, and the cells exhibited increased contractility, increased mature elastin production, and better mechanical properties than the static controls [12].

It is evident through the evaluation of some of these models that the bioactivity of the scaffold material, not just the material properties, significantly affects SMC behavior. The phenotype of SMCs seeded on purely synthetic matrices differs from that on a synthetic/natural mixture or just natural ECM components, and the benefits are generally seen with the addition of natural matrices. For example, TGFβ2-eluting scaffolds composed of PCL and gelatin significantly altered SMC numbers and cell areas on seeded scaffolds by changing the ratio of PCL to gelatin in combination with TGFβ2 dose [65]. Meanwhile, vascular grafts, fabricated via co-electrospinning collagen and hyaluronic acid and implanted in rats, enhanced the host SMC regeneration and proliferation into the graft [66]. In another study, a PCL/collagen I scaffold seeded with endothelial progenitor cells (EPCs) and SMCs, and implanted in sheep, exhibited extensive engraftment to the host vasculature and produced vast amounts of collagen, elastin, and GAGs in a structure resembling native ECM [67]. 

The most common natural ECM component to use as a scaffold for tissue-engineered small diameter vascular grafts is collagen, but fibrin, elastin, chitosan, and silk are also commonly used, in addition to decellularized blood vessels from animals [61]. In the collagen scaffolds, the fibrillar collagen density has been shown to significantly modulate the secretory factors and cytokines released by iPSC-derived SMCs, with higher densities resulting in the creation of a hypoxic pro-angiogenic and anti-inflammatory microenvironment [68]. In tubular nanofiber collagen scaffolds laden with iPSC-derived ECs internally and iPSC-derived SMCs externally, the collagen fiber alignment prompted cell alignment and consequently reduced the inflammatory response [69]. Meanwhile, in tubular collagen gels with SMCs only, the addition of cyclic strain induced an increased contractile marker expression and mechanical strength [70]. 

To better facilitate cell incorporation and growth in scaffolds, bioprinting has also been used as an in vitro blood vessel fabrication method, utilizing various combinations of natural and synthetic components and chemical modifications of these components. This has resulted in successful long-term cell viability (120 days) [71], mature cellular behavior [72], and recapitulation of some of the material properties (such as the storage modulus) similar to that of in vivo measurements [73]. In one example, tubular coaxial printing of patient-derived ECM encapsulating ECs or SMCs induced the successful and extensive endogenous deposition of collagen I and some deposition of elastin after 2-week culture with pulsatile flow through the lumen [74]. Separate from bioprinting but in another whole-cloth construction methodology, tissue-engineered rings of iPSC-derived SMCs have also been formed directly from a single cell suspension using clever ring-shaped culture dishes [75], and Raman spectroscopic analysis enabled highly specific characterization of iPSC-derived SMC phenotypic shift in response to growth media formulation [76].

## 4. Engineering of Smooth Muscle to Elucidate Mechanisms of Vascular Disease

Various tissue engineered constructs with human cells have already been applied to model certain vascular diseases to understand the underlying mechanisms of disease progression better. These models can also be used to identify therapeutic targets and to test potential mitigation or treatment strategies in the human mimicry models, thus potentially increasing successful clinical translation.

### 4.1. Hypertension, Pulmonary Arterial Hypertension, and Atherosclerosis

Tissue-engineered models have led to a better understanding of some of the mechanisms involved in vessel wall remodeling, and they have been used to model hypertension and pulmonary arterial hypertension (PAH). Both of these life-threatening pathologies are generally defined by characteristic vessel-wall stiffening and medial thickening in arterial vasculature induced by dysfunctional SMCs and fibroblasts [77,78]. While hypertension is a very common systemic condition usually resulting from external factors or aging and has a myriad of treatment options, PAH is an often incurable chronic condition in which the lung vasculature becomes mechanically obstructed, and the specific cause or disease mechanism is unknown for many patients [79]. Thus, it is important to be able to model both of these conditions in different ways in vitro. In one model of hypertension, explanted rabbit aortas were cultured under perfusion and showed activation of ERK 1/2 signaling cascade via FAK activation under high intraluminal steady stretch, while pulsatile stretch initiated the ERK1/2 signaling without FAK activation [80]. Meanwhile, PAH has been modeled in an artery-on-a-chip device using patient-derived cells. The characteristic PAH phenotype, including intimal thickening and arterial remodeling seen in vivo, was successfully demonstrated in the in vitro model [81]. 

Atherosclerosis, another common life-threatening cardiovascular condition, is usually brought on by external factors and aging, similarly to hypertension. Yet, the pathology is defined by the formation of intimal plaques that obstruct blood flow. These form due to inflammatory activation of ECs and lipid and immune cell accumulation, and dysfunctional SMCs contribute to plaque formation and progression in several ways [82]. The mechanisms of SMC involvement in atherosclerosis have been elucidated using 3D tissue-engineered models. In one example, a tissue-engineered “flap” laden with SMCs was subjected to complex pulsatile flow to model pathological ECM production as seen in the atherosclerotic plaques, and investigators found that doxycycline treatment abrogated the effects of the complex flow [83]. In another study, the atherosclerotic plaques were modeled with SMCs cultured in collagen gels in calcifying media, revealing that the extracellular vesicles secreted by SMCs in these conditions were calcified, thus further contributing to the plaque formation [84].

### 4.2. Insult and Acute Injury

Invasive procedures, such as plaque removal or vascular graft implantation, can induce acute vascular injury. To model this type of injury and work toward mitigating potential outcomes of restenosis or neointimal hyperplasia in patients, a bioreactor system was developed to culture ex vivo rat and human vessels and subjected them to a controlled injury that vigorously removed the luminal ECs from the vessels. The researchers found that SMC proliferation and neointima formation was reduced by seeding the injured vessels with human umbilical vein ECs prior to subjecting the vessels to simulated arterial flow in a bioreactor. Importantly, this result correlates with the clinical outcomes after vascular graft implantation in animals and humans [85]. Other insults investigated using tissue engineering techniques include radiation and oxidative stress. In one study, reactive oxygen species production induced by the treatment of isolated arterioles with lysophosphatidic acid resulted in significant changes in SMC contractile behavior in response to intraluminal flow when the arterioles were cultured in a bioreactor system [86]. In addition to these examples, a variety of acute insults including specific environmental factors and pharmaceuticals could be readily modeled with similar tissue engineered systems.

### 4.3. Genetic Mutations

Genetic mutations can seriously inhibit the ability of SMCs to secrete ECM and maintain their contractile function properly, which can result in lethal cardiovascular dysfunction in patients [87]. Tissue engineered models can provide deeper insight into the mechanisms through which these genetic mutations manifest vascular dysfunction.

For example, human SMCs from either healthy patients or those affected by genetic mutations resulting in abdominal aortic aneurysms (AAA) were isolated and cultured on poly-lactide-co-glycolide scaffolds for up to 5 weeks, and the differences in cytoskeletal alignment and ECM production were elucidated [88]. Progeria syndrome, a lethal vascular disease resulting from a mutation in the Lamin A gene, was modeled using iPSCs reprogrammed from a patient with progeria. The iPSC-SMCs were then seeded in tubular collagen scaffolds and cultured with ECs under peristaltic flow. The constructs exhibited increased medial thickness, calcification, apoptosis, and diminished vaso-activity, recapitulating the progeria pathology [89]. Another genetic disease associated with SMCs and ECM dysregulation, Marfan syndrome, was modeled using SMCs derived from iPSCs from a Marfan patient. These SMCs were subjected to cyclic stretch and exhibited decreased contractility, upregulated TGFβ signaling, and increased ECM accumulation: all characteristics of the syndrome [90].

## 5. Conclusions and Future Directions

Tissue engineering approaches have shown that SMCs respond differently to various stressors when in a 3D multicellular environment with mechanical stimulation, rather than the traditional 2D static culture systems. Additionally, further mechanistic insight into specific disease phenotypes has been achieved with human cells in tissue engineered in vitro models. A downfall of many of the successful tissue engineered models is that they utilize primary, explanted patient blood vessels or cells, which can inhibit the ability to achieve high throughput and clinically relevant experiments. A promising solution to this is to combine the wide variety of 3D designs and mechanical/chemical stimulation with the use of iPSC-derived SMCs from patients to minimize sample variability and provide a readily available cell source [6,51,75,89,90,91]. One aspect that could be explored more with these 3D models, particularly in the case of small diameter vascular grafts, is the use of natural matrix materials that better encompass the composition and structure of native ECM. The utilization of these materials has already been shown to induce more native-like behavior in cultured SMCs, as described previously. Further exploration of these materials in vascular grafts could facilitate more clinically relevant models and more insight into CVD mechanisms than is currently possible [11,72]. In addition, computational modeling before designing 3D vascular systems allows researchers to implement better design principles and forego much of the costly empirical experimentation usually needed to optimize these vascular systems [92,93,94]. Ultimately, combining all of these techniques can move the field forward in analyzing cardiovascular disease mechanisms in more depth and further testing potential future therapeutics or mitigation strategies that could help patients. is a review article

## Figures and Tables

**Figure 1 bioengineering-09-00449-f001:**
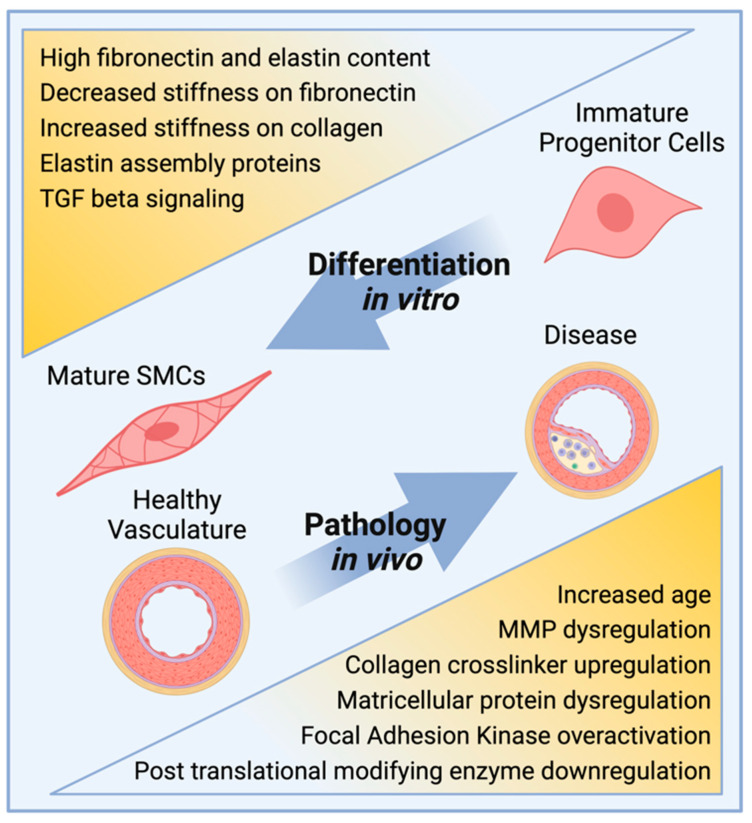
Schematic illustration of factors that prompt SMC maturation in culture systems (top) or disease progression in animal models (bottom). Created with BioRender.com.

**Figure 2 bioengineering-09-00449-f002:**
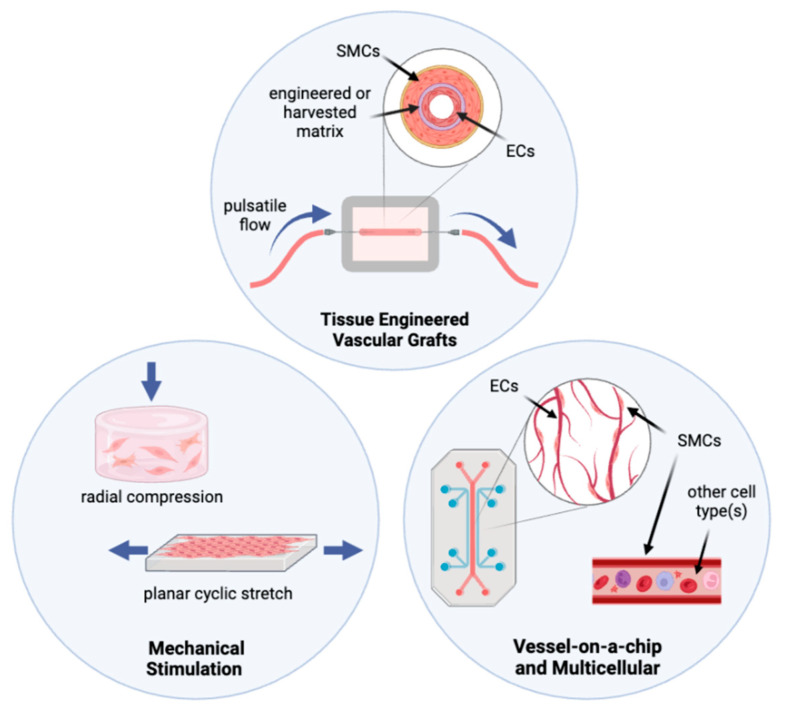
Schematic illustration of smooth muscle tissue engineering approaches. (**top**) Tissue Engineered Vascular Grafts, (**left**) Mechanical Stimulation Systems, and (**right**) Vessel-on-a-chip and multicellular culture systems. Created with BioRender.com.

**Table 1 bioengineering-09-00449-t001:** Summary of recent studies covering ECM factors involved in SMC behavior regulation.

Extracellular Matrix (ECM) Property	Factor or Pathway of Interest	Experimental Model	Findings
**Stiffness**	Lysyl oxidase-like 2 (LOXL2)	Human smooth muscle cells (SMCs) in 2D culture; LOXL2 knockout mice	LOXL2 promotes vascular stiffening by increasing overall cell and matrix stiffness and SMC contractility [16]
	Mineralocorticoid receptor (MR)	MR deleted male mice	Progression of cardiac fibrosis is mitigated by MR deletion Therapeutic antagonism of MR produced antifibrotic biomarkers [17]
	Circulating molecules	Parabiosis with young and old mice	In young mice, introduction of blood from old mice upregulated genes related to pathologic wall remodeling [18]
	Focal adhesion kinases (FAKs) and N-cadherin	FAK knockout mice	Inhibition of FAK activity blocks SMC proliferation and neointimal hyperplasia after injury [9]
		FAK and N-cadherin knockout mice	N-cadherin, in response to FAK activation, mediates cell–cell adhesion and SMC proliferation rate [19]
	Transforming factor beta (TGF-β) signaling pathway	Human SMCs on collagen I (COL1)-coated polyacrylamide (PA) gels	SMC contractile phenotype is induced as substrate stiffness increased; Inhibition of TGF-beta receptor reversed the stiffness effects [8]
		Human SMCs on polymethylsiloxane (PDMS)	SMCs expressed highest levels of osteogenic markers on intermediate stiffness gels (0.9 MPa) as opposed to high or low stiffness [20]
		Human SMCs on silk fibroin gels	Softer gels induced maturation of mesenchymal stem cells into SMCs [21]
	DNA methyltransferase I (DNMT1)	Human SMCs on fibronectin (FN)-coated PA gels; acute aortic injury and chronic kidney failure mouse models	Substrate stiffening induced synthetic phenotype in SMCs DNMT1 is repressed in stiffening arteries of both mouse models DNMT1 inhibition facilitates increased arterial stiffening in mice, and cellular stiffening and calcification in vitro [22]
	Ascending thoracic aortic aneurysm	Healthy and aneurysmal human SMCs on compliant hydrogels	Cytoskeletal stiffness was increased as substrate stiffness increased; aneurysmal cells exhibited increased traction forces compared to healthy cells [23]
**Fibrillar** **Protein** **Composition**	Fibronectin	Porcine SMCs suspended in COL1-FN gels	Fibronectin promoted elastin deposition and expression of assembly proteins; gel contraction and elastic modulus were increased in fibronectin-laden gels [24]
	Elastin	Human smooth muscle cells on porous collagen-elastin scaffold sheets	Elastin promoted mechanical and viscoelastic properties similar to native vessels, and contractile SMC phenotype [25]
	Collagen 1 and fibronectin	Human smooth muscle cells on ECM-coated polyacrylamide gels	SMCs on COL1-coated gels showed decreased migration and increased stress fiber orientation, and more organized cytoskeleton on stiffer gels, while the reverse was true for FN-coated gels [26]
**Non-fibrillar Protein Abundance and Structure**	Advanced Glycation End products (AGEs)	Mice models	AGEs increase vascular stiffness by prompting collagen crosslinking and inflammatory activation [27]
	Hyaluronic acid (HA)	Human SMCs cultured on micropatterned and HA/ECM-coated titanium	HA/ECM surface inhibits excessive SMC proliferation [28]
	Rho-related BTB domain–containing protein 1 (RhoBTB1)	Angiotensin-II treated (hypertensive) mice	RhoBTB1 alleviates arterial stiffness via actin depolymerization, but does not reverse hypertension [29]
	Elastin assembly proteins	Knockout mice models	Genetic deletions of specific elastin polymerization proteins, such as fibulin-4, fibrillin-1, lysyl oxidase, etc. degrade the integrity of elastin-contractile units, resulting in a range of disease phenotypes [30]
	Small leucine-rich repeat proteoglycans	Human coronary artery bypass patients	High pulse wave velocity was associated with significant downregulation of these proteoglycans, implicating involvement of collagen fibrillogenesis [31]
	Lysyl hydroxylase I (PLOD1), lysyl oxidase (LOX)	Human and mouse SMCs cultured in osteogenic medium	LOX overexpressing mouse SMCs exhibited increased calcification and increased collagen crosslinking [32]
	Post-translationally modified (glycosylated) fibronectin (gFN)	Rat SMCs on fibronectin	SMC adhesion to glycosylated fibronectin (gFN) was increased compared to native fibronectin, and was integrin independent; RAGE inhibition blocked adhesion to gFN [33]
	Protein and lipid phosphatase (PTEN)	Mice with SMC-specific-PTEN knockout; Isolated human atherosclerotic arteries	PTEN expressed in the SMC nuclei regulates the Serum Response Factor, maintaining the contractile phenotype; and PTEN expression is decreased in human atherosclerotic lesions [34]
	Matrix metalloproteinase-12 (MMP12)	MMP12 knockout mice	Deletion of MMP12 abrogates arterial stiffening by reducing elastin degradation [35]
	Matrixmetalloproteinase-9 (MMP9)	Macrophage depleted mice	Resident macrophages regulate collagen production in SMCs by MMP9 production, mediated by interaction of macrophages with hyaluronan [36]

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
