# Peer review of "Engineering Smooth Muscle to Understand Extracellular Matrix Remodeling and Vascular Disease"

_bioengineering, 2022, doi:10.3390/bioengineering9090449_

Round 1
Reviewer 1 Report
This is a review article regarding smooth muscle cells and extracellular matrix. The authors discussed recent studies using traditional in vitro culture systems and animal models that have improved the understanding of how specific properties of the ECM in the vasculature modulate SMC phenotype and how the SMCs in turn, alter the makeup of that ECM, and how other external cues like flow, mechanical strain, and biochemical signals are used to create three-dimensional (3D) tissue-engineered in vitro models of vascular tissue. The authors proposed that in vitro engineering of human vascular smooth muscle can facilitate a better understanding of relevant cardiovascular diseases using high throughput experiments, thus potentially leading to therapeutics or treatments to be tested in the future.
This reviewer considers that the authors have well written this review article. This reviewer has some comments as described below.
Major comments:
1. The authors mainly discussed the role of ECM using in vitro system, regarding stiffness, fibrillar protein composition, non-fibrillar proteins and matrix modifiers, in which several molecules play important roles. For better understandings for readers, the author should put graphical summary in this article.
2. Lines 314-323. Vascular disease part. Hypertension and pulmonary arterial hypertension (PAH) are different. Thus, line 314 should be “4.1 Hypertension, pulmonary arterial hypertension, and atherosclerosis”.
3. If the role of ECM is different from systemic arteries from pulmonary arteries, the authors should add this issue in this review article.
4. The authors should check the reference style.
Author Response
This is a review article regarding smooth muscle cells and extracellular matrix. The authors discussed recent studies using traditional in vitro culture systems and animal models that have improved the understanding of how specific properties of the ECM in the vasculature modulate SMC phenotype and how the SMCs in turn, alter the makeup of that ECM, and how other external cues like flow, mechanical strain, and biochemical signals are used to create three-dimensional (3D) tissue-engineered in vitro models of vascular tissue. The authors proposed that in vitro engineering of human vascular smooth muscle can facilitate a better understanding of relevant cardiovascular diseases using high throughput experiments, thus potentially leading to therapeutics or treatments to be tested in the future.
This reviewer considers that the authors have well written this review article. This reviewer has some comments as described below.
Major comments:
- The authors mainly discussed the role of ECM using in vitro system, regarding stiffness, fibrillar protein composition, non-fibrillar proteins and matrix modifiers, in which several molecules play important roles. For better understandings for readers, the author should put graphical summary in this article.
As suggested by the reviewer, we generated a graphical summary (new Figure 1) of all the aspects of the ECM that we review. Table 1 provides a summary of all the accompanying molecules/pathways.
- Lines 314-323. Vascular disease part. Hypertension and pulmonary arterial hypertension (PAH) are different. Thus, line 314 should be “4.1 Hypertension, pulmonary arterial hypertension, and atherosclerosis”.
We thank the reviewer for pointing this out. We have corrected the title.
- If the role of ECM is different from systemic arteries from pulmonary arteries, the authors should add this issue in this review article.
As suggested, we clarified the difference in the two conditions in the revised manuscript as following:
Tissue-engineered models have led to a better understanding of some mechanisms involved in vessel wall remodeling, and they’ve been used to model hypertension and pulmonary arterial hypertension (PAH). Both of these life-threatening pathologies are generally defined by characteristic vessel wall stiffening and medial thickening in arterial vasculature induced by dysfunctional SMCs and fibroblasts [78, 79]. While hypertension is a very common systemic condition usually resulting from external factors or aging and has a myriad of treatment options, PAH is an often incurable chronic condition in which lung vasculature becomes mechanically obstructed, and the specific cause or disease mechanism is unknown for many patients[80]. Thus, it is important to be able to model both conditions in different ways in vitro. In one model of hypertension, explanted rabbit aortas were cultured under perfusion and showed activation of ERK 1/2 signaling cascade via FAK activation under high intraluminal steady stretch, while pulsatile stretch initiated the ERK1/2 signaling without FAK activation [81].
And
Atherosclerosis, another common life-threatening cardiovascular condition, is usually brought on by external factors and aging, similarly to hypertension. Still, the pathology is defined by the formation of intimal plaques that obstruct blood flow. These form due to inflammatory activation of ECs and lipid and immune cell accumulation, and dysfunctional SMCs contribute to plaque formation and progression in several ways[83]. Mechanisms of SMC involvement in atherosclerosis have been elucidated using 3D tissue-engineered models.
- The authors should check the reference style.
Corrected

Reviewer 2 Report
In this review the authors describe the application of bioengineering approaches to facilitate in vitro study of vascular smooth muscle cell phenotype in health and disease. The review is comprehensive bringing together a large array of work from across the field and is excellently written and well presented. I very much enjoyed reading this piece and I feel other researchers would be interested too. Apart from a few very minor typos, I can find nothing else to critique about this piece.
(Very) minor comments
P4, L81; p5, L;104 p8, L254 – I would suggest altering ‘bulk’ to ‘mechanical’ as this does not feel the best word for the description used.
`p4, L87 'mineral corticoid' should be 'mineralocorticoid'
P4, L89 - an extra space is included at the start of the sentence.
P5, L101 and L107 – extra spaces to be removed
P8, L244, Additional ‘-‘ used
P9, L273 and 275. The β symbol appears to be missing in the PDF version of the article.
Author Response
We thank the reviewer for their positive feedback. All minor issues have been corrected in the reviewed manuscript.
Reviewer 3 Report
The article is very important, not existing much articles about vascular smooth muscle cells and their effects and importance. The tissue engineering approaches, namely the 3D multicellular systems are an important advance. This article represents a good review of what has been done in the last years in the area.
However, I think it would be important to add the planar cell surface area (PCSA) technique which is a primordial cellular tool to analyze the contractility of these cells, as well as to talk a little about the Patch Clamp technique, that is used to measure the activity of calcium, potassium currents.
Author Response
We thank the reviewer for their comments. We have added both to the revised manuscript:
In Ln 77 we added” When SMCs are cultured on 2D substrates, detailed phenotypic characteristics can be evaluated through a variety of methods. Contractile marker expression can be quantified as protein or RNA transcript [8, 9]. The degree of functional contractility can be determined via planar cell surface area (PCSA) measurement before and after treatment of cells with vasodilating or vasoconstricting agents [10-12]. Calcium and potassium ion channel activity, which directly induces SMC contractile function, can be evaluated via electrical current measurement across the cell membrane using patch clamp electrophysiology [13-15].”
In Ln 257 we added “Maturation of mesenchymal cells into SMC-like cells has been achieved through co-culture with fibroblasts in decellularized ECM; this maturation was confirmed using PCSA analysis [11].”
Round 2
Reviewer 1 Report
This reviewer has no further comment.
Reviewer 3 Report
The authors have revised the manuscript according to the reviewer's comments.